# Efficient WSN Node Placement by Coupling KNN Machine Learning for Signal Estimations and I-HBIA Metaheuristic Algorithm for Node Position Optimization

**DOI:** 10.3390/s22249927

**Published:** 2022-12-16

**Authors:** Bastien Poggi, Chabi Babatounde, Evelyne Vittori, Thierry Antoine-Santoni

**Affiliations:** 1UMR CNRS 6134 SPE, University of Corsica, 20250 Corte, France; 2UAR CNRS 3514 Stella Mare, University of Corsica, 20250 Corte, France; 3UMR CNRS 6240 LISA, University of Corsica, 20250 Corte, France

**Keywords:** wireless sensor network, deployment, optimization, KNN, HBIA

## Abstract

Wireless sensor network (WSN) deployment is an intensive field of research. In this paper, we propose a novel approach based on machine learning (ML) and metaheuristics (MH) for supporting decision-makers during the deployment process. We suggest optimizing node positions by introducing a new hybridized version of the “Hitchcock bird-inspired algorithm” (HBIA) metaheuristic algorithm that we named “Intensified-Hitchcock bird-inspired algorithm” (I-HBIA). During the optimization process, our fitness function focuses on received signal maximization between nodes and antennas. Signal estimations are provided by the machine learning “K Nearest Neighbors” (KNN) algorithm working with real measured data. To highlight our contribution, we compare the performances of the canonical HBIA algorithm and our I-HBIA algorithm on classical optimization benchmarks. We then evaluate the accuracy of signal predictions by the KNN algorithm on different maps. Finally, we couple KNN and I-HBIA to provide efficient deployment propositions according to actual measured signal on areas of interest.

## 1. Introduction

In the past two decades, the number of deployed wireless sensors has increased dramatically. This spectacular growth is on the one hand correlated with the development of machine learning applications requiring large amounts of data and on the other hand made possible by the reduction in the production costs of the equipment.

Following this trend, many new technologies have emerged supporting different routing protocols, different operating systems, and different micro-controller boards. They have been successfully applied in several fields of research, and application such as environmental monitoring [1,2], smart agriculture [3], health [4], and security [5].

This current heterogeneity of technologies generates new challenges [6]. To deal with it, researchers have to provide new tools to help decision-makers during WSN deployment. Many issues need to be addressed: “Which technology is the most suitable for my problem?”; “How to configure WSN parameters?”; “Where to deploy sensors in order to observe a specific phenomenon?”; “How to ensure node connectivity within the studied area according to its specificities and constraints?”.

One of the most encountered problems with WSN is their efficient placement in an area of interest (AoI). On star topology, all sensors must receive signals from one or more fixed antennas. Many times these antennas are placed in specific places accepted by the population. Signal propagation is rarely the main criterion considered. Citizens usually focus first on other criteria, such as visual impact and health protection. Node localization must reconcile these constraints and provide a minimum level of connectivity.

## 2. Wireless Sensor Network Deployment

### 2.1. WSN Deployment Problematic

A WSN generally consists of two groups of entities: the gateways and the sensors. The gateways, or “sink nodes”, are usually antennas to which the collected data converge. They do not collect physical data but rather act as a relay between the WSN and other larger networks such as the Internet or an organization’s intranet. The sensors are composed of four parts: the acquisition unit, which allows physical collection of the observed phenomena; the calculation unit, which allows the sensor’s operating system and applications to function; the power supply unit, which provides energy to the various components; and the transmission unit, which allows the data to be transferred to the gateway or to other relay nodes. These devices will measure physical values and transmit them directly or by hops to the gateways.

WSN deployment quality is a highly subjective concept [7] depending on several factors: the application field, the integrated software technologies, the hardware technologies, the legislative constraints, as well as the access constraints to the deployment areas. The chosen deployment strategy can indirectly optimize different features of a WSN, including network coverage, network connectivity, network robustness, and network reactivity. Node localization is also an active field of research for WSN, which includes three groups of approaches: “range based algorithms” [8,9,10], “range free algorithms” [11,12,13], and “hybrid algorithms” [14]. The first group is based on signal physical information analysis such as time of arrival (TOA) or received signal strength indicator (RSSI). The second group is based on neighborhood analysis, with several methods such as DV-hop, APIT, or sequence-based algorithms. The third group is a combination of the first two. All these methods focus on network connectivity.

Several definitions of connectivity can be found in the literature [15,16,17], depending on the network topology used. Received and transmitted signal estimations are generally based on the calculation of Euclidean distances between nodes. For mesh topologies, the minimum number of hops between nodes to communicate with gateways is estimated and neighborhood analysis is performed. For star topologies (i.e., LoRa technology), the strongest received signal from different antennas describes connectivity quality.

Currently, two families of deployment strategies can be found in the literature: the “classical” approaches based on geometry and the “stigmergic” ones based on guided randomness.

#### 2.1.1. Classical Approaches

The problem of deploying wireless sensor networks has emerged following the spread of this technology. The first deployment assistance tools were based on three kinds of methods. The first one was based on deployment grids [18,19]. The second relied on the concepts of attraction and repulsion forces between the nodes of the network [20,21], and the third one was based on “computational geometry” with Delaunay triangulation and Voronoi diagrams [22,23]. An example of such a deployment is shown in Figure 1.

Although these approaches have been frequently used successfully, they have some disadvantages. In return for their fast execution time, they are based on a theoretical approximation of the signal, making it challenging to consider the deployment zone’s specificities and dynamic aspects. Moreover, their adaptation to the particularities of the studied problem requires critical adaptation times and in-depth knowledge of the deployment zone. In other cases, these specificities are simply ignored, negatively impacting the proposed scenarios’ feasibility. Faced with these important limitations, their use is becoming less and less common in the literature. New methods based on guided random reasoning bypass these limitations.

#### 2.1.2. Modern Approaches

WSNs are a recent technology with many research directions [6]. The deployment of a WSN is an NP-complete problem [24]. Indeed, the geographical position of the sensors will influence many parameters of the WSN, such as its lifetime, fault tolerance, speed of detection of studied phenomena, as well as the accuracy of the measurements. There is no exact algorithm to solve all these objectives in a global and confident way.

Nevertheless, in order to meet this need for “intelligent deployments”, optimization methods known as “metaheuristics” are regularly used to respond to the problems encountered by the WSN [25].

These methods are a set of nature-inspired algorithms for minimizing or maximizing the parameters of an objective function (i.e., maximizing network coverage). They are regularly used with success in the field of WSN. Examples include localizing captured phenomena [26,27,28,29], creating and configuring routing protocols [30,31,32,33,34,35], or creating clusters to maximize network lifetime while minimizing energy consumption [36,37,38,39,40,41,42,43,44,45].

Due to their high genericity and adaptability, they are also regularly used for optimizing WSN deployments [46,47,48,49,50,51,52,53,54,55,56,57,58,59,60].

These modern, iterative optimization methods are based on two central concepts: “intensification” and “diversification”. They start with a random generation of solutions in the search space under study, followed by a guided search according to the results provided for an evaluation function representing the answer to the studied problem. Several hundred so-called “metaheuristic” methods are available in the literature today, as well as numerous variants. This phenomenon generates a certain complexity in the choice of the method for a given problem [61]. Moreover, their performance depends on many factors, such as the quality of the algorithm implementation, parameterization, the random generator used, and solution representation. There are many sources of inspiration for these methods, such as evolutionary theory [62,63], swarms [64,65,66,67,68], physics [69,70,71], or human behavior [72,73,74]. Examples of historically older metaheuristics are given in Table 1, and more recent methods are shown in Table 2.

These optimization methods are not guaranteed to find the optimal deployment but an acceptable deployment in a reasonable time, in contrast to classical optimization methods whose computation time on such problems would exceed a thousand years. The main advantage of this method is the rapid integration of the different criteria for estimating the deployment quality into the evaluation function of the problem. An example of such a deployment is shown in Figure 2 with the evaluation function of maximizing area coverage. In this figure, the maximum coverage is not reached, but the proposed solution is very close to it.

Therefore, a fundamental question arises to guarantee the success of the proposed deployment: which evaluation function to choose? In the case of our work, this evaluation function will be strongly conditioned by the signal quality between sensors and gateways. However, this signal quality is difficult to estimate by simulations. Accurate measurements are generally necessary, but these are rarely exhaustive. It is, therefore, essential to propose an approach that combines the deployment’s optimization with an accurate signal estimation based on measurements collected in situ.

## 3. Proposed Approach

### 3.1. Coupling a Metaheuristic with a KNN Evalution Function

Our method is based on two basic concepts. The first one is the optimization of the sensor position by using a metaheuristic. The second one is an evaluation function based on signal quality estimations at the chosen location. This estimation is based on the KNN algorithm and real measures from the area of interest. This operation is explained in a simplified way in Figure 3.

In the example in Figure 3, we can see different signal qualities collected by field measurements. We can see that the antenna’s signal is lower to the northeast of the antenna than to the southeast of the antenna, probably due to obstacles. If we want to place a node on the area of interest and ensure that it is connected to the antenna, its position will be the one represented by the yellow diamond. This process is repeated on different areas of the map. We observe that it is possible to meet the dual objective of “collecting data in a specific area” (sensing) and “routing the data to the gateway” (transmitting).

### 3.2. Signal Estimation with KNN Algorithm

Many machine learning algorithms are available in the literature today. They can be split into two groups: “supervised algorithms” for classification and regression and “unsupervised algorithms” for clustering and dimensionality reduction. Optimization methods require a precise evaluation function to be minimized or maximized according to the problem to be solved. In our approach, we chose to use machine learning to “predict” by regression the quality of a signal on a studied area. Different supervised learning algorithms can be used for that purpose [78,79]. Most of these algorithms, such as neural networks (NN), require a training process. Moreover, they are challenging to explain [80], and many parameters need a tuning process [81].

However, some are simpler to implement and provide acceptable short calculation time. The KNN algorithm is a “lasy learning” machine learning method [82,83]. This algorithm estimates a point’s value according to its neighboring by computing average distances between instances, as shown in Figure 4 and Equation (Equation 1). We assume this concept is suitable for signal estimation on geolocated nodes inside a deployment area. For example, if neighbors B and C of node A have a good-quality signal, we can assume that node A will also have a good-quality signal without considering all nodes inside the deployment area.
(1)$$estimated\_signal_{x}=\frac{\left(\sum_{i=1}^{K}Neighbor_{i},x\right)}{K}$$

### 3.3. Optimize Node Location with I-HBIA Metaheuristic

#### 3.3.1. Canonical HBIA Algorithm

In our approach, we have chosen to use the HBIA algorithm [84]. The behavior of the birds in Alfred Hitchcock’s famous film *The Birds* inspires this recent metaheuristic. As stated in the “no free lunch” theorem, predicting which metaheuristic will perform best on a given problem is impossible [61]. Nevertheless, there are three main reasons for the choice of this algorithm for our approach:This algorithm belongs to the family of P-Metaheuristics that base the optimization process on a set of solutions and are generally more efficient for optimization problems containing local minimums.This algorithm, in contrast to other metaheuristics, does not require many parameters. The only parameterization concerns the number of birds found in the colony. Moreover, the impact of the variations of this parameter on the results obtained remains marginal.This algorithm offers superior performance to many other metaheuristics on high dimensional problems. This aspect is particularly interesting for deploying sensor networks where the number of nodes and their positions can quickly become very large.

The intelligent behavior of birds during attacks inspired the authors of the HBIA algorithm. Each solution is represented as a bird in the colony. In the rest of this paper, we will use the following conventions: *B* for the whole bird colony, $B_{i}$ for a specific bird, and $B_{*}$ for the best bird.

As summarized in Figure 5, the algorithm starts with a random solution generation phase. A lurking phase is performed to avoid too-fast convergence of the algorithm towards a local minimum. It allows for the birds $B_{i}$ to move away from the best bird $B_{*}$, allowing the search to diversify.

Once the lurking phase is over, the birds will be evaluated in order to designate them to one of three categories: the best, the elites, and the others. Only one best bird is defined in the algorithm. The number of birds belonging to the elite is estimated according to the colony’s size. Once this distribution is complete, the birds will carry out successive waves of attacks. Two types of attack are possible depending on their relative distance from the best bird. The first will approach the best bird directly in the search area, while the second will take a less direct route. These attacks will allow birds to get progressively closer to the best bird without missing the diversification during the search. During optimization, a mechanism based on a dynamic parameterization of aggressiveness will ensure an efficient modulation between intensification and diversification. If a bird does not progress during optimization, its aggressiveness will increase. This aggressiveness will be correlated with the level of diversification of its search. On the other hand, if its evaluation improves, its aggressiveness will remain stable.

When a phase of “stagnation” is identified, a phase of reorganization of the colony starts. It will lead to the deletion of the solutions with the worst results. The best birds and the elite birds will survive. For the others, their probability of survival will be proportional to their evaluation, and the worse their rating, the greater their risk of dying. Dead birds will be removed from the colony and replaced by new birds. For this purpose, the algorithm will consider previously generated variables that led to better solutions than the previous ones.

This process is repeated until the maximum iteration number is reached or the evaluation function’s result exceeds a specified threshold. When the optimization process ends, the proposed deployment coordinates are saved or displayed to the decision-makers.

To efficiently apply the HBIA algorithm to the specificity of the search space of a WSN deployment, a modified version of the algorithm has been designed: I-HBIA (“Intensified-Hichcok Bird Optimization Algorithm”). This hybrid algorithm includes a local search process to decrease the number of optimization iterations and increase solution quality.

#### 3.3.2. The I-HBIA Algorithm

The optimization process by a metaheuristic presents some specificities compared to other optimization problems. The first step of a metaheuristic is a systematically random generation. Generally, this type of deployment offers acceptable solutions when the objective of the deployment is to maximize coverage. The sensors will be uniformly deployed over the area. The optimization process will therefore focus during the different iterations on refining a solution that is already correct from the first iteration.

In the classical version of the algorithm, the best bird does not evolve during iterations. In the I-HBIA algorithm, we have therefore chosen to add a step that runs in parallel with the different attacks: “cloning the best bird,” as described in Figure 6. This step consists of duplicating the best bird a number of times corresponding to the number of dimensions studied. For example, a problem of dimension 50 will generate the creation of 50 clones. These clones will be more or less faithful replicas of the best bird. Indeed, they will be subjected to random perturbations. Each clone will be assigned a mutation number corresponding to its replication number, as shown in Figure 7.

This hybrid version of the algorithm allows two things. It will enable faster convergence of the algorithm. It also allows us to get closer to the optimum, especially when the number of dimensions is large (more than 30 dimensions). The following section presents the different results obtained.

## 4. Results

### 4.1. KNN Algorithm Parametrization

KNN algorithm parametrization is a complex task generally based on empirical observations. The parameter K is the number of considered neighbors to estimate the signal quality. If the amount of measured data is limited, the algorithm’s performance can be volatile and challenging to measure, as shown in Figure 8. We can see in this figure that the prediction precision is variable when the algorithm works with 200 or 250 measured values. The volatility of results is less intense in scenarios where we have a large volume of data, such as 900 or 1000 measured values.

To find the best value of K, we test the algorithm performance on different maps with different measures. The various parameters used for this test are described in Table 3. As shown in Figure 9, the better performance is provided by K = 2. Other values of K provide acceptable results but need more computation time to compute the neighborhood.

### 4.2. Signal Estimations on Maps

To evaluate the performance of the KNN algorithm for signal quality estimation, we have generated different signal maps. These signal maps are of two types: “without obstacles” and “with obstacles”. Maps estimate the circular signal around the antennas with a linear signal strength attenuation. These signal representations can easily be updated in our architecture to consider the specificities of the hardware used in a given deployment.

#### 4.2.1. Signal Map without Obstacles

Figure 10 demonstrates the functioning of the KNN algorithm with a parameterization of K = 2. On this map, we can easily observe that the quality of the accuracy is strongly correlated to the number of available measurements. We can observe a high accuracy from 200 measures (NM = 200).

Once we validated signal estimation on signal measurements without obstacles, we choose to test our approach on maps containing antenna signal disturbances.

#### 4.2.2. Signal Map with Obstacles

On the map shown in Figure 11, we also observe that the KNN algorithm performs well for estimating signals. We also notice that the amount of measurement data also conditions the accuracy of the predicted signals. This aspect is even more critical because some interfered areas might not be detected without a sufficiently large number of measurements. This aspect would lead to wrong signal quality predictions. For example, we can see in the figure that signal perturbations are not correctly estimated in the upper right corner of the map (NM = 10 to NM = 200). These perturbations are only estimated from a considerable number of measurements (NM = 500 and NM = 1000).

The different tests presented above confirm the high adaptability of the KNN algorithm to the dataset, and the obstacles are well taken into account by the algorithm. We now have a reliable estimation function to predict the signal received at a specific point in the deployment zone. This function will be used as an evaluation function during the execution of the optimization process by the I-HBIA algorithm.

#### 4.2.3. Influence of the Number of Measurements on the Quality of Predictions

We run different estimations on a dataset with various maps to accurately estimate the number of “real signal quality measurements” needed to perform accurate “signal quality estimations”.

Each configuration (number of measurements) was tested on 100 different signal maps. The results shown on the curve in Figure 12 logically confirm the importance of a large number of real signal data for the accuracy of the estimated signals.

We can conclude that 400 measured signals provide an accurate signal estimation for the KNN algorithm. However, we must balance this aspect. Prediction precision can be influenced positively and negatively by the homogeneity of the measured signals. In our maps, signals are measured homogeneously, and there are examples of deployment where this is impossible.

Before optimizing the nodes’ positions, we chose to verify the performance of the I-HBIA algorithm on classical benchmark functions. We also compared the performance of our version of the HBIA algorithm (I-HBIA) to the classical version (HBIA).

### 4.3. I-HBIA Algorithm Performances

Each optimization process was repeated ten times, and four configurations were tested. These are described in the Table 4.

We evaluated the performances of these two algorithms on unimodal benchmark functions, presented in Table 5, and multimodal benchmark functions, shown in Table 6. The evaluation space of the unimodal and multimodal functions with two dimensions can be seen in the 3D graphs in Figure 13.

The evolution of the best solution on unimodal and multimodal functions is presented in Figure 14 and Figure 15. These figures show that the four configurations converge towards the global minimum of the functions. We can observe from these average results that the I-HBIA algorithm converges systematically faster towards the global minimum. The hybrid aspect of this algorithm can explain this and the settlement principle presented previously. This improvement in performances is particularly significant for the following functions: $U_{3}$, $U_{4}$, $M_{2}$, and $M_{3}$. The results between the different methods are close on other functions, but the I-HBIA algorithm remains superior ($U_{1}$, $U_{2}$, $U_{5}$, $U_{6}$, $M_{1}$, $M_{5}$, $M_{6}$). The function $M_{4}$ is the only function for which the results of the classical version of the algorithm are better.

However, these results must be nuanced because this algorithm version requires more calls to the evaluation function at each iteration due to cloning the best solution step. Nevertheless, the intensification of the search allows a more refined precision of the best solution obtained, which will not be negligible for a WSN deployment problem for which a slight variation in the quality of the connectivity can have consequences on the whole network.

When the I-HBIA algorithm’s effectiveness was validated on different benchmarks, we used the algorithm on deployment problems.

### 4.4. KNN and I-HBIA for WSN Deployment

Following the validation of the signal estimation by the KNN algorithm and the proof of the I-HBIA optimization method, we tested our approach on the different optimization scenarios described in the Table 7.

We can observe three phenomena on the deployment examples proposed by the tool and presented in Figure 16 and Figure 17. In the first case, the zone of interest does not contain any antenna. In this case, the sensor will try to get as close as possible to a boundary of its zone on which the estimated signal will be the highest. In the second case, the area of interest contains only one single antenna. In this case, the sensor will place itself as expected, close to the antenna, where the estimated signal will be the highest. Finally, in the third case, the area contains several antennas. In this case, the sensor will move closer to the antenna where the estimated received signal will be maximum, provided that no obstacle is present on any of the antennas in the area. If an obstacle is present and generates signal disturbances, the sensor will find the position where it receives the strongest signal from one of the antennas in the area.

However, as we can see in Figure 16d and Figure 17d, the deployment optimization does not consistently achieve the optimum. Indeed, in these figures, we can see that a zone does not contain any sensor while another contains two. This phenomenon is characteristic of metaheuristics. In this case, the optimization has made good progress but has finally been blocked in a local minimum. Generally, the repetition of optimization processes brings a solution to this kind of problem, even with a smaller number of iterations. This problem is no longer visible when the optimization is performed on a more significant number of iterations on the same number of zones.

As shown in Figure 18, the method also works on many dimensions. On this map, we optimize a deployment on 25 zones. The problem to solve is dimension 50 (latitude and longitude of each node). We can observe from this figure that only one zone of the 25 defined does not contain any sensor. On the areas covered by a sensor, we can see that each sensor chooses the position at which its signal estimation is maximum.

Nevertheless, we can observe from this figure that a few sensors are outside areas of interest. Three factors can explain this. The first one concerns the coverage of the zone by signals. Indeed, if any signal does not cover this zone, the optimizer will not place any sensor in this zone. The second factor is the size of the area. A small area will be problematic for the optimizer, and the guidance towards this area by the evaluation function will be weak. The third factor is the number of iterations used for this execution. Indeed, 500 iterations for a complex problem of dimension 50 have a high probability not of leading to an optimum but getting very close to it.

## 5. Conclusions

Our approach provides a new method to optimize the deployment of WSN in signal estimation functions. The core of this approach is based on two algorithms. The KNN algorithm accurately estimates the signals on the deployment area from localized points. The I-HBIA algorithm optimizes the nodes’ positioning according to the estimated signals and the defined area of interest.

This work opens the way to different research tracks. The first one concerns the accuracy of signal measurements. Currently, our method works on simulated signal collections on other test datasets. In the future, it would be interesting to integrate signal measurement data collected by drones or technicians traveling to the deployment areas.

During our simulations, we realized that the computation time remained important despite using metaheuristics. Indeed, the geometrical operations necessary to evaluate the solutions and search for the neighbors for the estimations require considerable computation time. Solutions might include discretizing the search space while maintaining an acceptable accuracy level and continue improving the I-HBIA method by creating a parallel version of I-HBIA.

During some optimization processes, I-HBIA does not find the best node positions. This aspect increases with the number of deployed sensors and can be explained by the curse of dimensionality. To address this limitation, we will integrate reduction methods into our approach.

Finally, we will focus on weather impact on the LoRa signal quality. Indeed, during our experimentations, we observed some variations in signal quality measures, and we assume this variation was generated by environmental factors (rain, wind, or snow). This aspect can be hazardous for critical applications such as animal tracking or water level thresholds monitoring.

## Figures and Tables

**Figure 1 sensors-22-09927-f001:**
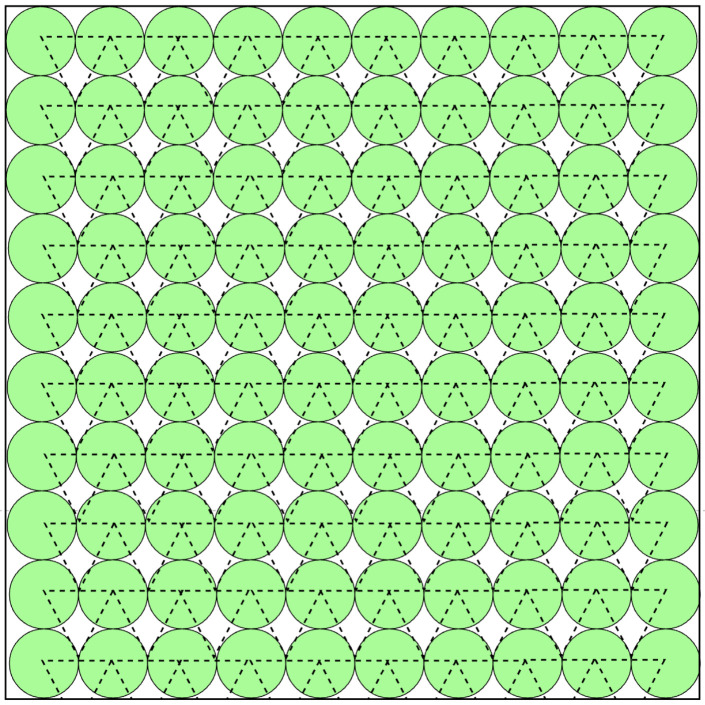
An example of a classical deployment strategy.

**Figure 2 sensors-22-09927-f002:**
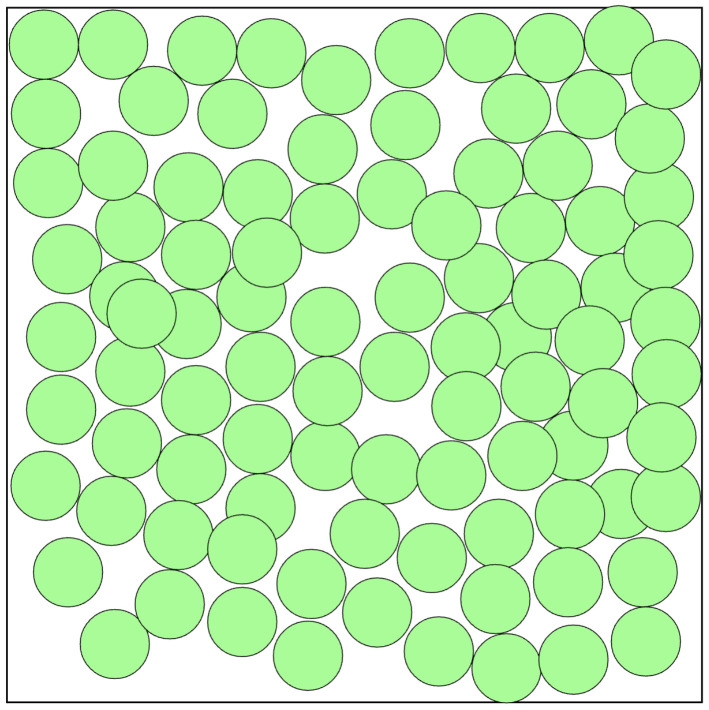
Example of random deployment strategy.

**Figure 3 sensors-22-09927-f003:**
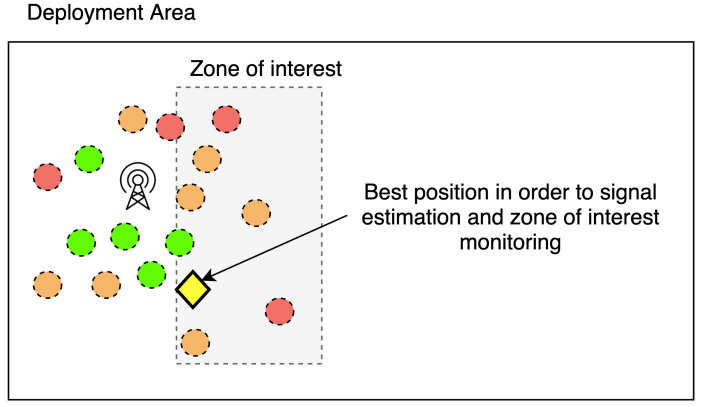
Example of optimal proposed position for a node.

**Figure 4 sensors-22-09927-f004:**
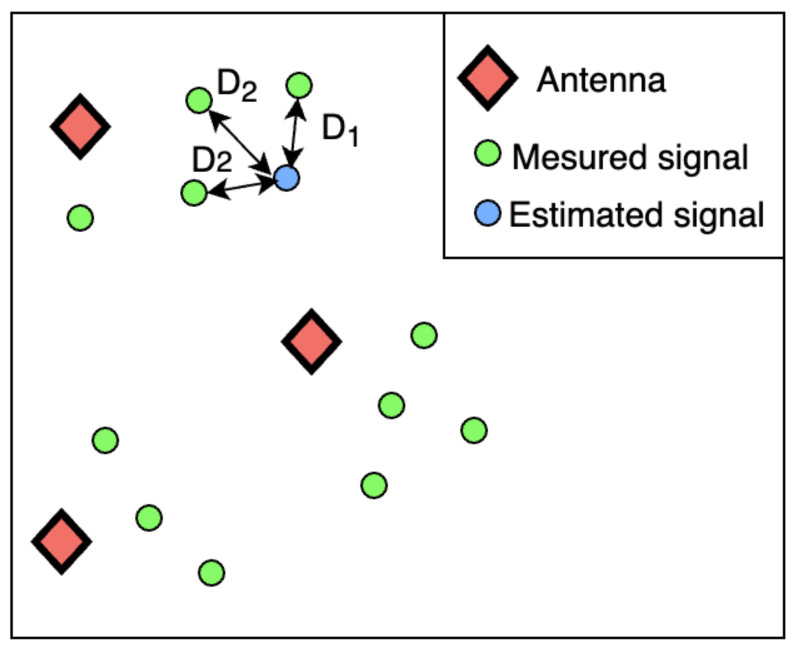
Signal estimation with KNN algorithm (K = 3).

**Figure 5 sensors-22-09927-f005:**
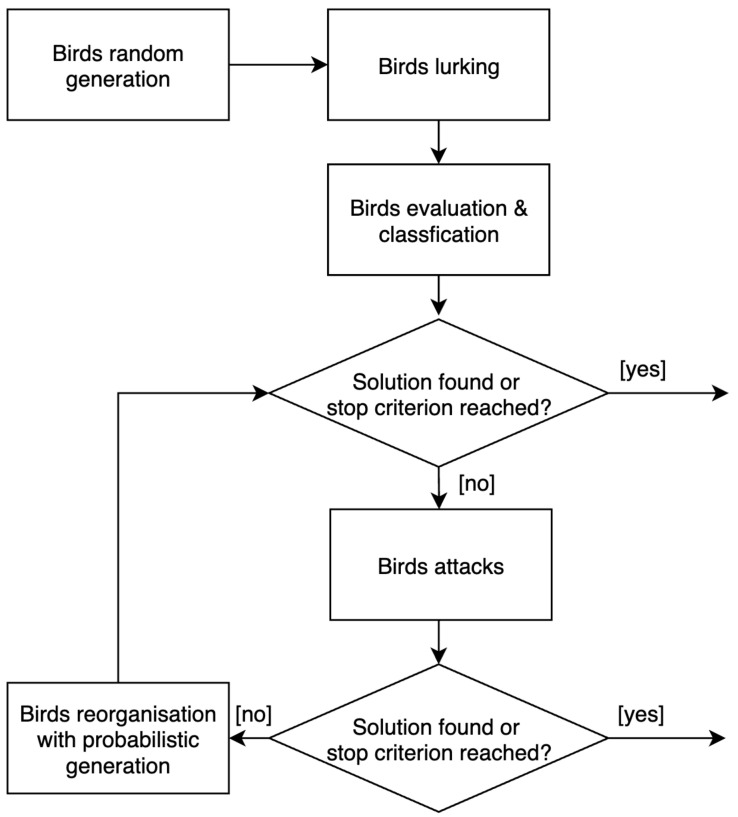
The different steps of the HBIA optimization algorithm.

**Figure 6 sensors-22-09927-f006:**
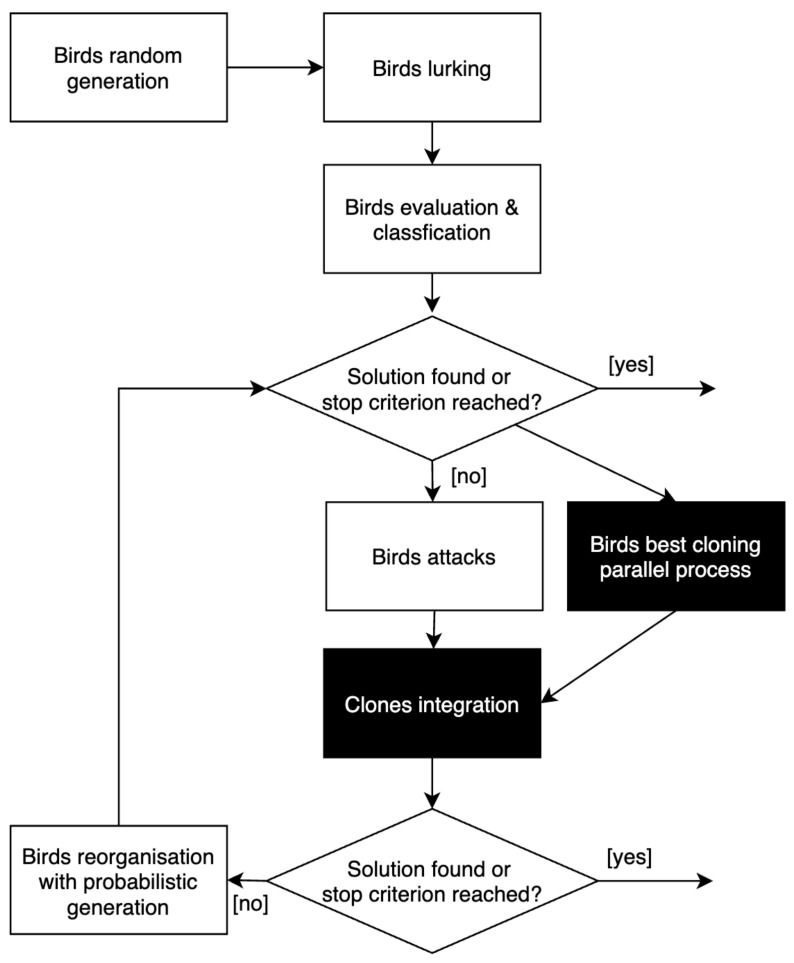
The different steps of I-HBIA hybrid optimization algorithm.

**Figure 7 sensors-22-09927-f007:**
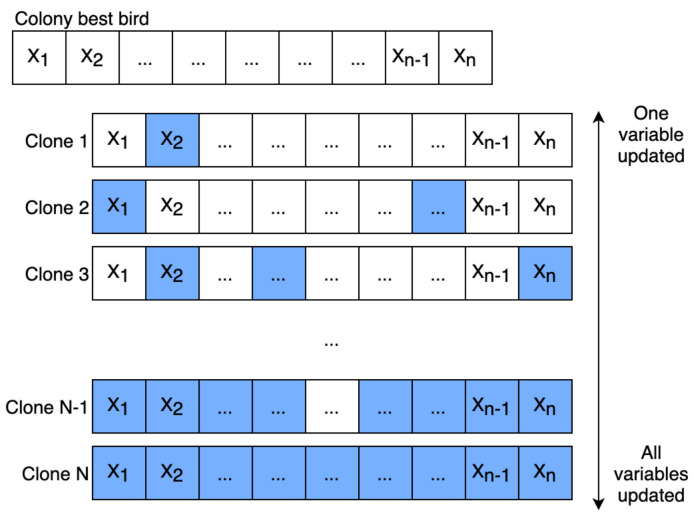
Intensified research step in I-HBIA.

**Figure 8 sensors-22-09927-f008:**
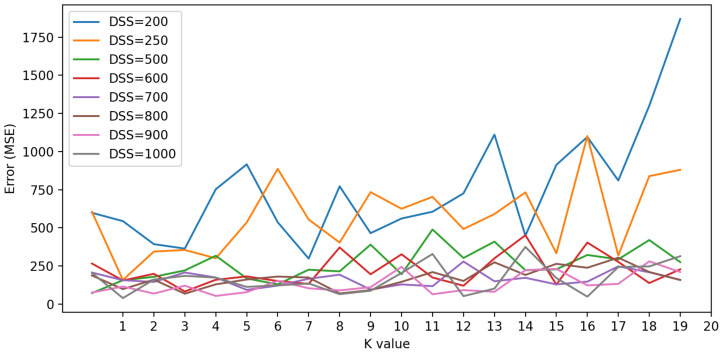
Errors for different dataset size (DSS) with different K values.

**Figure 9 sensors-22-09927-f009:**
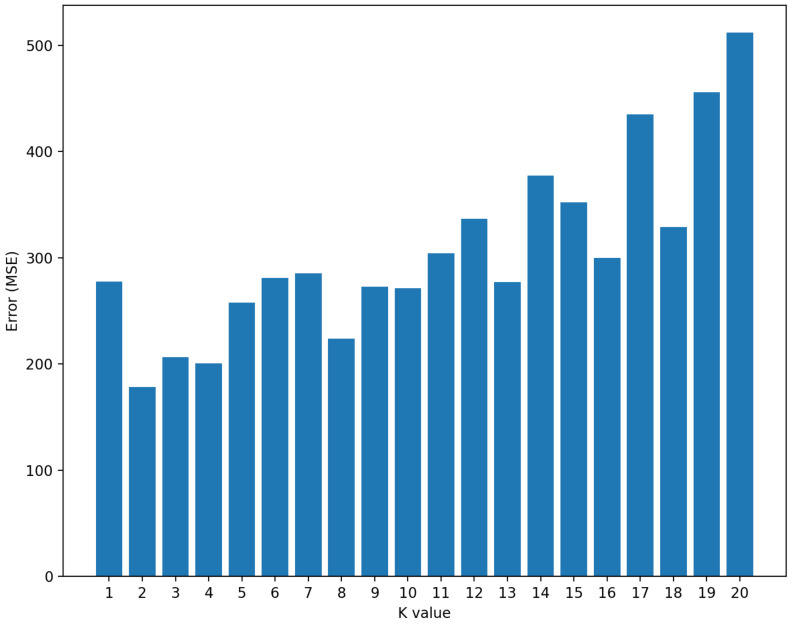
Estimated signal prediction error with different K values.

**Figure 10 sensors-22-09927-f010:**
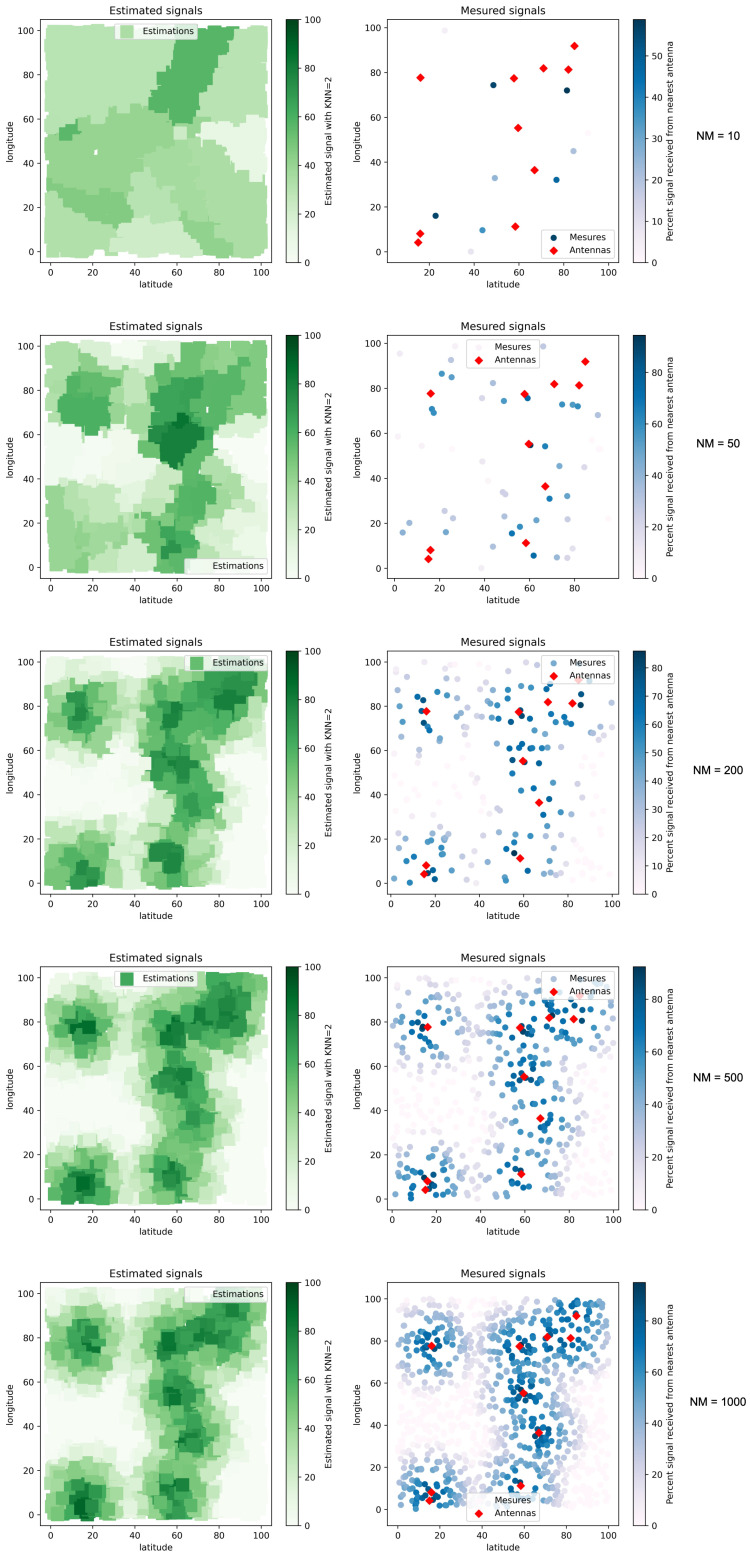
Signal estimations with different numbers of signal measures (NM).

**Figure 11 sensors-22-09927-f011:**
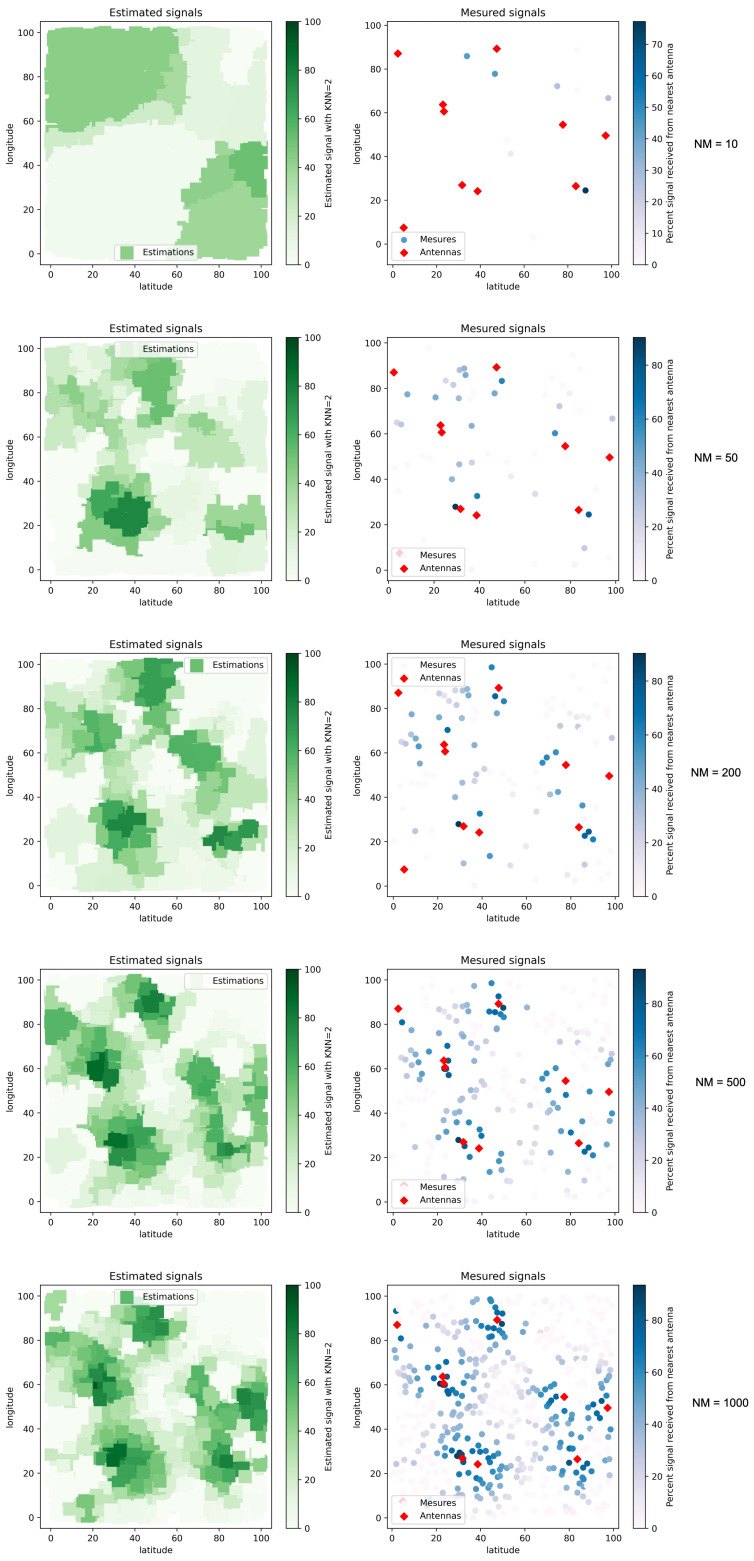
Signal estimations with different numbers of signal (NM) measures and obstacles.

**Figure 12 sensors-22-09927-f012:**
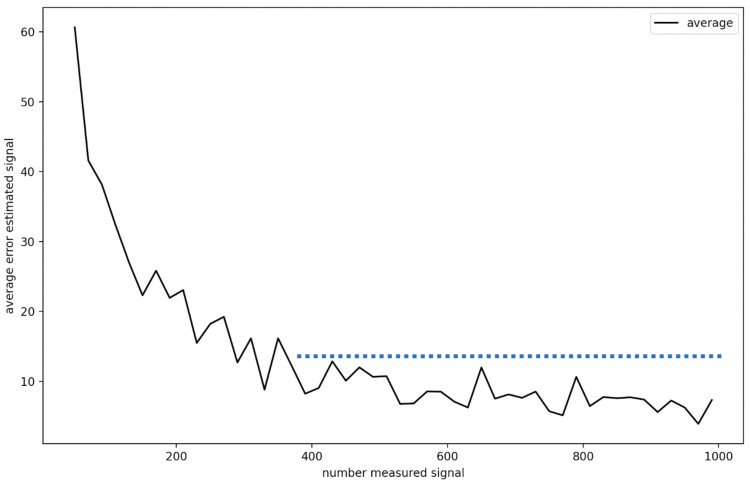
Influence of the number of measured signals on the estimated signal accuracy.

**Figure 13 sensors-22-09927-f013:**
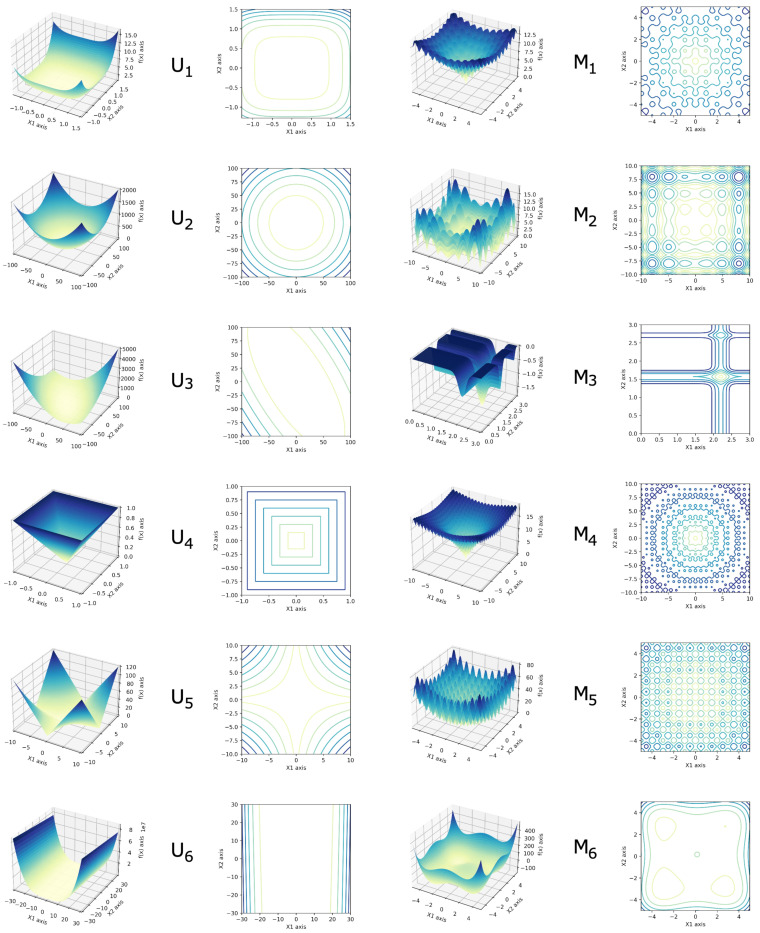
Three-dimensional plot of two dimension function representation.

**Figure 14 sensors-22-09927-f014:**
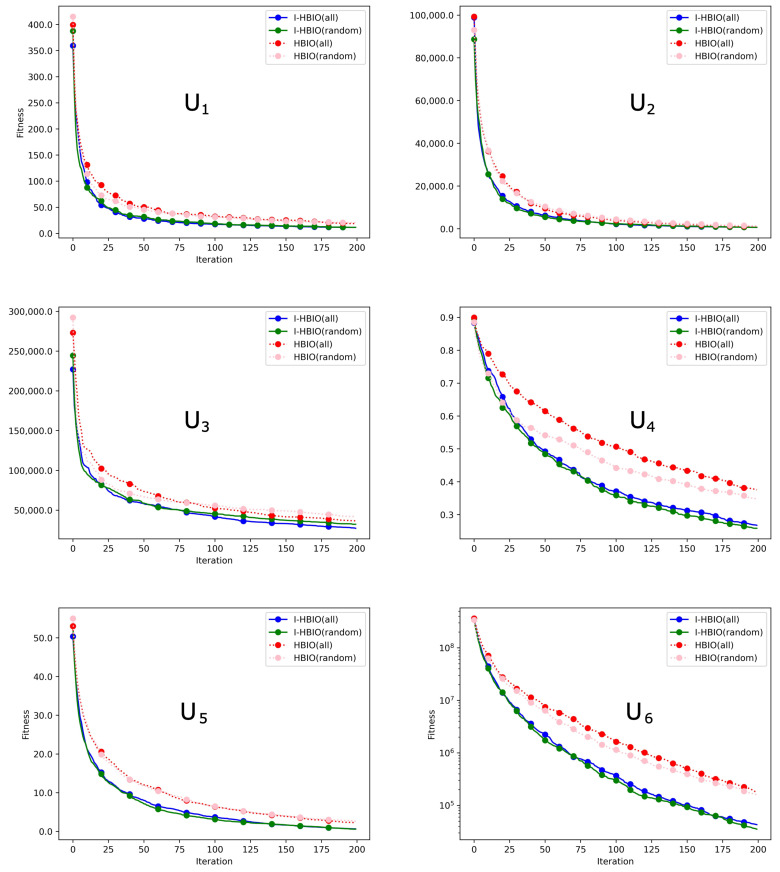
Optimization results on unimodal benchmark functions.

**Figure 15 sensors-22-09927-f015:**
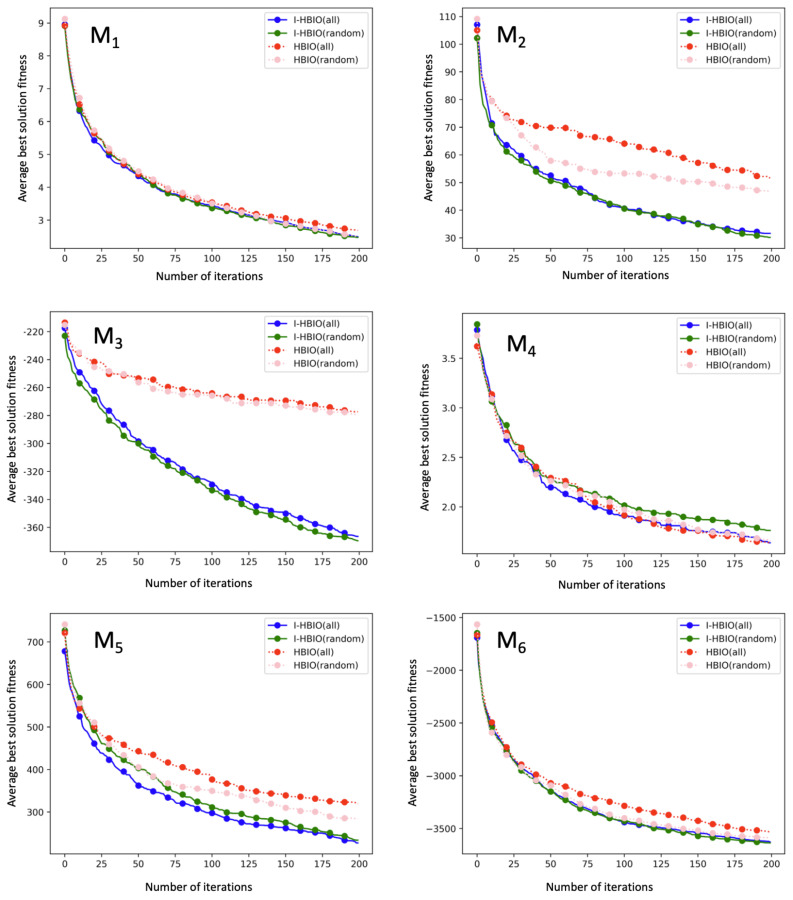
Optimization results on multimodal benchmark functions.

**Figure 16 sensors-22-09927-f016:**
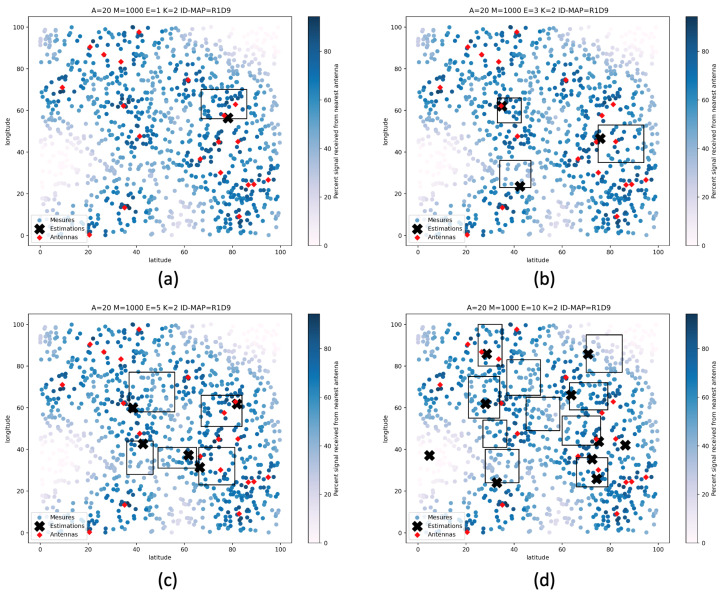
Proposed deployments after 50 iterations on 1, 3, 5, and 10 areas of interest without obstacles (**a**) 1 area of interest, (**b**) 3 areas of interest, (**c**) 5 areas of interest, (**d**) 10 areas of interest.

**Figure 17 sensors-22-09927-f017:**
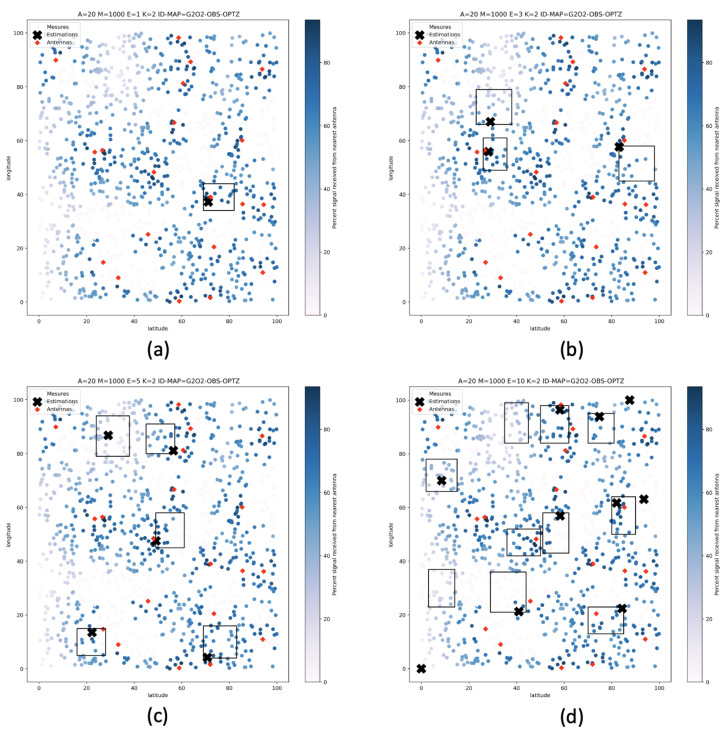
Proposed deployments after 200 iterations on 1, 3, 5, and 10 areas of interest with many obstacles (**a**) 1 area of interest, (**b**) 3 areas of interest, (**c**) 5 areas of interest, (**d**) 10 areas of interest.

**Figure 18 sensors-22-09927-f018:**
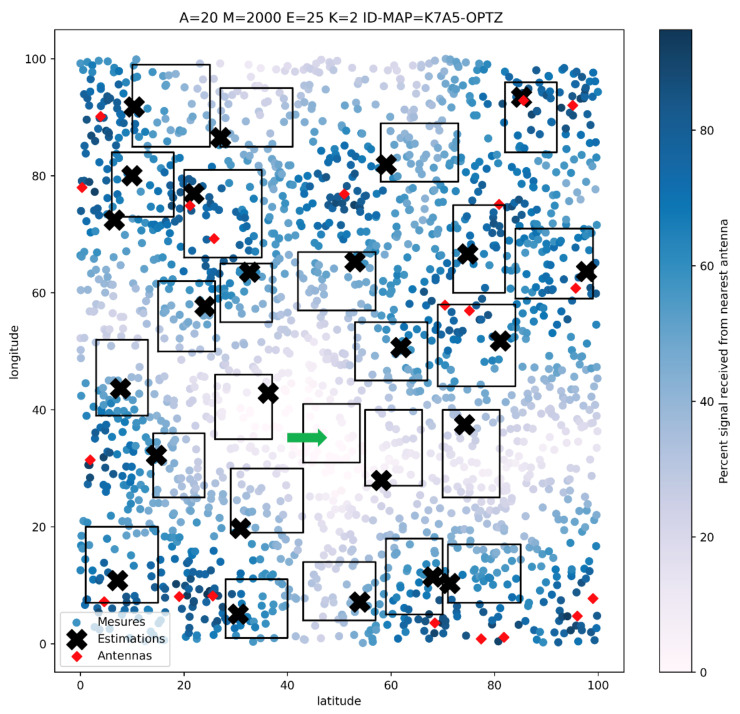
An example of proposed deployment for 25 areas of interest after 500 iterations.

**Table 1 sensors-22-09927-t001:** Some most used metaheuristics.

Name	Publication Date	Inspiration
Simulated annealing [75]	1983	Physics
Differential evolution [62]	1997	Evolutionary
Particle swarm optimization [65]	1995	Swarm
Genetic algorithms [63]	1962	Evolutionary
Ant colony optimization [64]	2006	Swarm

**Table 2 sensors-22-09927-t002:** Some promising recent metaheuristics.

Name	Publication Date	Inspiration
Grey wolves optimizer [66]	2014	Swarm
Whale optimization algorithm [67]	2016	Swarm
Ant lion optimization [76]	2015	Swarm
Moth-flame optimization [68]	2015	Swarm
Sine cosine algorithm [77]	2016	Mathematics

**Table 3 sensors-22-09927-t003:** Parameters used to find best K value.

Parameters	Values
Number of measures	[200, 250, 500, 600, 700, 800, 900, 1000]
K values	[1–20]
Number of repetitions	20
Considering learning dataset	80%
Considering testing dataset	20%

**Table 4 sensors-22-09927-t004:** Used configurations.

Configuration	Local Search by Cloning	Dimension Consideration	Number of Iterations
HBIA (all)	Disabled	All dimensions	200
HBIA (random)	Disabled	Random dimensions	200
I-HBIA (all)	Enabled	All dimensions	200
I-HBIA (random)	Enabled	Random dimensions	200

**Table 5 sensors-22-09927-t005:** Unimodal benchmark functions used.

Equation	Name	Dimension	Number of Iterations	Benchmark ID
$\sum_{i=1}^{n}x_{i}^{4}+random\left[0,1\right])$	Quartic	50	200	$U_{1}$
$\sum_{i=1}^{n}x_{i}^{2}$	Sphere	50	200	$U_{2}$
$\sum_{i=1}^{n}{\left(\sum_{i=1}^{n}x_{j}\right)}^{2}$	Schwefel 1.22	50	200	$U_{3}$
$\max_{i=1..n}\left|x_{i}\right|$	Schwefel 2.21	50	200	$U_{4}$
$\sum_{i=1}^{n}\left|x_{i}\right|+\prod_{i=1}^{n}\left|x_{i}\right|$	Schwefel 2.22	50	200	$U_{5}$
$\sum_{i=1}^{n-1}\left[100{\left(x_{i+1}-x_{i}^{2}\right)}^{2}+{\left(x_{i}-1\right)}^{2}\right]$	Rosenbrock	50	200	$U_{6}$

**Table 6 sensors-22-09927-t006:** Multimodal benchmark functions used.

Equation	Nane	Dimension	Iterations	Benchmark ID
$20+e-20\exp\left[-0.2{\sqrt{\frac{1}{n}\sum_{n}^{i=1}}}^{}x_{i}^{2}\right]-exp\left[\frac{1}{n}\sum_{i=i}^{n}\cos\left(2\pi x_{i}\right))\right]$	Ackley	50	200	$M_{1}$
$\sum_{i=1}^{n}\left|x_{i}\sin x_{i}+0.1x_{i}\right|$	Alpine	50	200	$M_{2}$
$\sum_{i=1}^{n}\sin\left(x_{i}\right){\left[\sin\left(\frac{jx_{i}^{2}}{\pi }\right)\right]}^{2m},m=10$	Michalewicz	50	200	$M_{3}$
$1-\cos\left(2\pi \sqrt{\sum_{i=1}^{n}x_{i}^{2}}\right)+0.1\sqrt{\sum_{i=1}^{n}x_{i}^{2}}$	Salomon	50	200	$M_{4}$
$10n+\sum_{i=1}^{n}\left(x_{i}^{2}-10cos\left(2\pi x_{i}\right)\right)$	Rastringin	50	200	$M_{5}$
$\frac{1}{2}\sum_{i=1}^{n}\left(x_{i}^{4}-16x_{i}^{2}+5x_{i}\right)$	Styblinski Tank	50	200	$M_{6}$

**Table 7 sensors-22-09927-t007:** Used configurations.

Number of Measured Signals	Signal Perturbations	Number of Areas of Interest	Number of Iterations	Figure Example
1000	no	1	50	Figure 16a
1000	no	3	50	Figure 16b
1000	no	5	50	Figure 16c
1000	no	10	50	Figure 16d
1000	yes	1	50	Figure 17a
1000	yes	3	50	Figure 17b
1000	yes	5	50	Figure 17c
1000	yes	10	50	Figure 17d

## Data Availability

Not applicable.

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
