# Peer review of "Efficient WSN Node Placement by Coupling KNN Machine Learning for Signal Estimations and I-HBIA Metaheuristic Algorithm for Node Position Optimization"

_sensors, 2022, doi:10.3390/s22249927_

Round 1

Reviewer 1 Report

This paper proposes a novel large scale wireless sensor network deployment approach based on machine learning and a metaheuristic. In order to prove the effectiveness of the algorithm, the KNN algorithm is applied to the measured data. This method is innovative to some extent, but some problems need to be addressed as follows:

1.     In fig. 6, what actions will the algorithm perform after finding the solution or reaching the stopping criterion?

2.      In section 4.1.1 signal map without obstacles, what features of the estimated signal showing the relationship between quality of the accuracies and the number of measuring devices available?

3.      What does Figure example mean in Table 6? Why are there fields in the table but no records?

4.      Some related studies on node placement and selection in WSN for positioning are recommended to be surveyed and compared, e.g., Device-to-device cooperative positioning via matrix completion and anchor selection; A novel range-free node localization method for wireless sensor networks.

5.      In the experimental results section, fig. 11 and fig. 13 lack horizontal and vertical coordinates. In particular, performance differences between different methods cannot be seen in some figures. It is suggested to add local map to make the result clearer.

Author Response

Dear colleague,

Thank you for your helpful feedback. Please find attached our response to your comments.

Best regards,

Reviewer 2 Report

Authors present their proposition of a novel approach to detect sensor node locations in large scale WSNs, based in a KNN machine learning algorithm and an improved metaheuristic model.

The paper has several english flaws, needing an extensive editing of the English language. Overall, it affects negatively the scientific soundness of the paper.

Although the subject is relevant and the approach is quite relevant for nowadays WSN applications, the authors must perform a paper reformulation.

Many orthographic and grammatical errors are present throughout the paper. Some examples:

Line 4 - "nammed"

Line 17 - "environement"

Line 20 - "guarante"

Line 22 - "Witch"

Line 27 - "interrest"

Line 29 - "criteras"

Line 31 - "this constraints"

The above examples are found in the first page of the paper. Please consider a thorough review of the paper to correct these.

Other english problems are related to poorly constructed phrases. One example:

Lines 24-25 - "How to insure node connectivity in order to area of deployment specificities, constraints?"

Overall, such errors demonstrate that the paper is lacking rigor and must be reformulated.

Besides the English problems, the authors must also change the paper construction. Figures are mostly presented without any discussion of its contents. Standard procedures also include that figures are presented immediately before or after their reference in the text. For instance, figure 6 is introduced in page 4 and only displayed in page 7. Figure 4 is neither introduced nor discussed.

Further below, the metaheuristic HBIA is referred many times as HBIO...including in the sub-section 3.3.1 title.

Authors describe their choice for the KNN is a general manner, lacking references for statements such as "Others do not need large volumes of data to make predictions.". Also, authors should not make use of detached sentences as the previous one.

Still regarding KNN approach, in the Results section, the authors should consider to prove their choices with results for different values of K, even if it is just a short comparison chart.

For the remainder of the Results section, the authors present several figures (8,9, 10, 11,...) that lack discussion and introduction to the reader of what is contained in each of the figures.

Author Response

(The authors gave the same response as above.)

Reviewer 3 Report

This paper introduced a node deployment scheme based on KNN and metaheuristic for large scale wireless sensor network. The metaheuristic algorithm was used to generate the best node location. The fitness function in the algorithm was based on connectivity maximization between nodes and antennas. The KNN algorithm was used to estimate signal quality. There are some issues that need to be addressed by the authors.

1. English writing needs to be significantly improved. There are many grammar mistakes. For example, in the abstract, "we nammed" should be corrected to "named" , "signals quality" should be "signal quality" or the quality of signal, "interrest" should be "interest", "This aspects allows" should be "This aspect allows" or "These aspects allow", "propositions" should be "proposition". It is strongly recommended that the authors correct all the grammar mistakes in the whole article.

2. The algorithms proposed by the authors paid much attention on the connectivity between nodes in the network. In recent years, several efficient node positioning methods based on the connectivity have been proposed. For example, the following article optimized the node position by making full use of the connectivity. The title is Connectivity Based DV-Hop Localization for Internet of Things, published in ieee transactions on vehicular technology. In the introduction, the difference between the connectivity in the above article and this paper is suggested to be briefly introduced.

3. Why KNN was used for signal estimation? The authors are suggested to explain their choice, since there exist many clustering algorithms including KNN.

Author Response

(The authors gave the same response as above.)

Round 2

Reviewer 2 Report

I am satisfied with most of the changes that the authors promoted on the previous version. However, I urge the authors to perform a  final thorough revision of the text to correct typos and poorly constructed sentences.

I provide some examples below:

Line 9 - "deployement"

Line 34 - "deployement"

Figure 5 and 6 - "Solution finded"

Line 364 - "We observed that the optimum could not be reached in many cases..."

Author Response

Dear colleague,

The whole article has been carefully revised to improve the English.
We have improved the individual sentences as much as possible.

If the level of English in this version is still insufficient, we can consider a correction by native English, but we will need more time.

Kind regards,